# ITS2 Sequencing and Targeted Meta-Proteomics of Infant Gut Mycobiome Reveal the Functional Role of *Rhodotorula* sp. during Atopic Dermatitis Manifestation

**DOI:** 10.3390/jof7090748

**Published:** 2021-09-13

**Authors:** Kevin Mok, Narissara Suratanon, Sittiruk Roytrakul, Sawanya Charoenlappanit, Preecha Patumcharoenpol, Pantipa Chatchatee, Wanwipa Vongsangnak, Massalin Nakphaichit

**Affiliations:** 1Department of Biotechnology, Faculty of Agro-Industry, Kasetsart University, Bangkok 10900, Thailand; kevin.m@ku.th; 2Specialized Research Unit: Probiotics and Prebiotics for Health, Faculty of Agro-Industry, Kasetsart University, Bangkok 10900, Thailand; 3Pediatric Allergy & Clinical Immunology Research Unit, Division of Allergy and Immunology, Department of Pediatrics, Faculty of Medicine, Chulalongkorn University, King Chulalongkorn Memorial Hospital, The Thai Red Cross Society, Bangkok 10330, Thailand; mayzped@gmail.com (N.S.); pantipa.c@chula.md (P.C.); 4Functional Ingredients and Food Innovation Research Group, National Center for Genetic Engineering and Biotechnology, National Science and Technology Development Agency, Pathum Thani 12120, Thailand; sittiruk@biotec.or.th (S.R.); sawanya2010@gmail.com (S.C.); 5Department of Zoology, Faculty of Science, Kasetsart University, Bangkok 10900, Thailand; preecha.pa@ku.th; 6Omics Center for Agriculture, Bioresources, Food, and Health, Kasetsart University (OmiKU), Bangkok 10900, Thailand

**Keywords:** mycobiome, meta-proteomics, atopic dermatitis, gut–skin axis, fungi, *Rhodotorula*

## Abstract

Association between the gut mycobiome and atopic dermatitis was investigated in 9–12-month-old infants using metagenomics. Two groups of atopic dermatitis infants were classified according to their symptom development as outgrown (recovered) and persisted (still undergoing). The evenness and diversity of the mycobiome in the persisted group were higher than in the healthy and outgrown groups. Dysbiosis of the microbiome in the persisted group was observed by a reduction in the *Ascomycota*/*Basidiomycota* ratio. Five fungi were selected as markers from each sample group. In the persisted group, *Rhodotorula* sp. abundance increased significantly, while *Wickerhamomyces* sp. and *Kodamaea* sp. abundance increased in the healthy group, and *Acremonium* sp. and *Rhizopus* sp. abundance increased considerably in the outgrown group. Metaproteomic analysis revealed that the persisted group had a high abundance of fungal proteins, particularly those from *Rhodotorula* sp. Unique proteins such as RAN-binding protein 1 and glycerol kinase from *Rhodotorula* sp. were hypothesized to be related to atopic dermatitis manifestation in infants.

## 1. Introduction

Atopic dermatitis is a recurrent, chronic skin inflammation that has a high prevalence in infants. It is thought to be a gateway of the atopic march, a sequence of allergic symptoms (i.e., food allergy, allergic rhinitis, and asthma) that often develops during infancy. Symptoms of atopic dermatitis typically occur in the first six months of life. Previous research [1,2,3] suggested a correlation between atopic dermatitis exposure and a higher chance of developing other symptoms later in life. Early atopic dermatitis diagnosis and treatment are critical to reducing the likelihood of atopic march progression. It is difficult to distinguish between atopic dermatitis and common skin allergies due to the contribution of its dynamic and multi-factor pathophysiology (e.g., skin barrier defects, immune response, and environmental factors) [4,5].

Several biomarker candidates along with gut microbiome have been implemented for early diagnosis of atopic dermatitis [6,7,8], with changes indicating an increased risk of certain diseases and vice versa [8,9,10]. The abundance of particular taxa is often used as an indicator for host health. A high *Firmicutes*/*Bacteroidetes* ratio was related to an increased risk of obesity [11], while *Fusobacteria* has been connected to an increased risk of colorectal cancer [12].

Association between the gut microbiome and atopic dermatitis was previously reported, with the growth of butyrate-producing bacteria positively correlated to milder symptoms [13] and reduction of *Faecalibacterium prausnitzii* linked with manifestation [14]. These findings improved the understanding of how gut microorganisms affect atopic dermatitis. However, because of the complex nature of the gut microbiota environment, further research on the relationship with other gut microorganisms such as fungi is required before definite conclusions can be drawn.

Fungi develop shortly after birth and become a fundamental component of the human gut [15,16]. Fungi share the same particular intestinal environment as bacteria [17], making their association vital in defining host health [18,19,20]. The study of gut fungi, also known as the gut mycobiome, has recently sparked much interest, especially regarding their roles in the development of inflammatory disorders [21]. Inflammatory bowel disease has been linked to a high prevalence of *Candida albicans* [22], while Chron’s disease has been linked to a high abundance of *Malassezia* [23]. Arrieta et al. [24] reported a more pronounced association between atopic wheeze and fungi compared to bacteria, mainly linked with *Pichia kudriavezii*. Previous research suggested that gut fungi play a significant role in disease progression. Fungi are known to play an essential role in the health of their hosts, but there is currently no agreement on what defines a normal gut mycobiome [20]. Further research into the gut mycobiome is needed to determine how fungi impact host health.

Metagenomic analysis has long been considered the gold standard in assessing gut microbiome profiles [25]. In Thai gut microbiome studies, Raethong et al. [26] sequenced the genes and annotated the metabolic functions of the gut microbiome in Thai adults using the whole metagenome shotgun (WMGS) approach. Identifying the core microbiome is an initial step to examine the relationship between gut constituents and their activities in sustaining the host’s health [27]. Moreover, metaproteomics-based analysis was further studied for the activity of microbial communities. Kingkaw et al. [28] recently investigated the microbial community composition of the human gut to identify different key proteins playing metabolic functional roles in the microbiome of healthy infants and infants with atopic dermatitis. However, the existence of microorganisms does not always reflect similarly in vivo metabolic characteristics [29]. Thus, a combined multi meta-omics approach is required to determine the gut microbiome function for a better understanding of complex interactions in the human gut, in particular disease regulation [30].

Here, the gut mycobiome (fungal microbiome) of infants with atopic dermatitis was investigated using both metagenomic and metaproteomic approaches to capture mycobiome status links to the disease. An explanation of the involvement of perceived mechanisms during atopic dermatitis manifestation was enhanced by functional metabolic analysis of proteins from each significantly different taxon. Furthermore, a quantitative real-time PCR (q-PCR) platform was used as an alternative detection method for the causes of atopic dermatitis.

## 2. Materials and Methods

### 2.1. Study Design

This study was part of a population-based birth cohort study conducted at King Chulalongkorn Memorial Hospital, Bangkok, Thailand. It focused on healthy infants and infants with atopic dermatitis who developed symptoms within the first twelve months of life. The infants were prospectively followed up until the age of 30 months. Early life environmental exposures, socioeconomic factors, parental health, and infant diet data were collected by questionnaires (Appendix A). Allergy diagnosis was confirmed by pediatric allergists. All infants included in this study met the American Academy of Dermatology criteria for atopic dermatitis [31], with the severity determined using the SCORAD score. A total of 34 infants were included in this study, comprising healthy (*n* = 17) and infants with atopic dermatitis manifestation (*n* = 17), sub-grouped into atopic dermatitis recovered (outgrown; *n* = 7) and still undergone atopic dermatitis symptoms (persisted; *n* = 10) at the time of sample collection. Stool samples were collected when the infants were 9–12 months old. Stool samples were immediately stored at −80 °C to prevent nucleic acid degradation. Informed consent from the infant guardians was obtained before their inclusion in this study, which was conducted according to the Helsinki Declaration, with the protocol approved by the Ethics Committee of King Chulalongkorn Memorial Hospital under approval reference number IRB-358/58.

### 2.2. Microbial DNA Extraction of Stool Samples

The microbial DNA extraction process was performed using the Qiagen QIAamp DNA stool kit (Qiagen, Hilden, Germany) according to the international human microbiome standard (IHMS) protocol [32]. In brief, the stool sample was resuspended (1:9 *w*/*v*) in phosphate-buffered saline (PBS) pH 8 and pelleted by centrifugation at 12,000× *g* for 10 min. The pellet was then homogenized with lysis buffer and transferred to a 2 mL tube containing 0.3 g of each sterile zirconia bead with 0.1 mm and 1 mm diameter (BioSpec, Bartlesville, OK, USA). Mechanical lysis was conducted using a FastPrep-24 benchtop instrument (MP Biomedicals, Santa Ana, CA, USA) at 6.5 m/s for 8 min with a series of one-minute beating and five-minute resting on ice. The extracted DNA quality and concentration were measured by a Nanodrop 2000c (Thermo Scientific, Waltham, MA, USA). After extraction, DNA samples were stored at −20 °C until required for further analysis.

### 2.3. Amplification and Sequencing of the Internal Transcribed Spacer 2 (ITS2) Region

The fungal ITS2 region was amplified from 150 ng/µL of microbial DNA by PCR using ITS3 (GCATCGATGAAGAACGCAGC) and ITS4-NGS (TCCTSCGCTTATTGATATGC) primer sets [33]. Each primer was integrated with the Illumina adapter and linker sequence at the 5′ end. PCR was performed using HotStarTaq DNA polymerase (Qiagen, Hilden, Germany) under the following conditions: pre-denaturation at 95 °C for 10 min; 30 amplification cycles of 95 °C for 20 s, 56 °C for 45 s, 72 °C for 1 min, and a final extension at 72 °C for 10 min. The positive control (10 ng/µL of *Saccharomyces cerevisiae* DNA) and negative control (nuclease-free water) were also included in each PCR run. The PCR product was visualized using 2% agarose, 90 V, 30 min and stained by ethidium bromide (EtBr). The PCR products were then purified using NucleoSpin Gel and PCR Clean-up (Macherey-Nagel, Düren, Germany). Samples were pooled and sequenced using the Illumina MiSeq Reagent V2 500 cycle (2 × 250 bp) kit on the Illumina MiSeq platform (Illumina, San Diego, CA, USA) according to the manufacturer’s instructions.

### 2.4. Mycobiome Determination from ITS2 Region Sequencing

All sequencing data were quality controlled and processed with BBDUK v38.86 [34] and USEARCH v10.0.240 [35]. Primer sequences were detected and trimmed with BBDUK. Then, each paired-end read was merged into a single sequence with USEARCH. Merged reads shorter than 150 bps, having more than 20 mismatches in the alignment, or having an expected error at more than 1.0 were rejected, and only the forward read was used. Operational taxonomic units (OTUs) were clustered with 97% threshold, and identification of representative sequences was performed by UPARSE pipeline with a cut-off value of 0.8 [36]. OTUs table was produced with USEARCH, and abundances were recovered by mapping OTUs table with the representative sequence. Taxonomy assignment was performed using the SINTAX-algorithm in USEARCH [37] with UNITE fungal ITS database v8.2 [38]. Alpha diversity was then calculated from the OTUs table with USEARCH. Primer 7 v7.0.20 was used to construct the Principal Component Analysis (PCA) of the beta diversity and a heat map displaying the abundance of fungus at the genus level. The beta diversity metric was calculated based on the relative abundance data at class level taxa using the Bray-Curtis distance. The raw Illumina sequences of ITS2 metagenome used in this study have been deposited at the NCBI shorts read archive (SRA) with the BioProject accession number PRJNA751473.

### 2.5. Quantitative Real-Time PCR (qPCR) Analysis of Five Distinct Genera

#### 2.5.1. Primer Design

Specific primer sets were designed according to their sequence of 18s rRNA gene for *Acremonium* sp. and 26s rRNA gene for *Rhodotorula* sp., *Wickerhamomyces* sp., *Rhizopus* sp. and *Kodamaea* sp.

Sequences of several species within their respective genera were retrieved from the NCBI database and aligned using the MUSCLE multiple sequence alignment algorithm [39] in MEGA 7. Multiple aligned sequences that represented the conserved region of each genus were chosen to run in PrimerQuest™ Tool (IDT, Coralville, IA, USA). The specificities of these primers were aligned with the NCBI Blast database (http://www.ncbi.nlm.nih.gov/BLAST, accessed on 7 October 2020). All primers used are listed in Table 1. Real-time techniques were applied to validate the primers with five target fungi, including *Acremonium* sp. TBRC-9800, *Rhodotorula mucilaginosa* TBRC-4420, *Wickerhamomyces anomalus* TBRC-651, *Rhizopus oligosorus* TBRC-BCC 17810, and *Kodamaea ohmeri* TBRC-3060. PCR product size and melting temperature were observed.

#### 2.5.2. Plasmid Construction and Standard Curve Creation

Genomic DNA of five fungi was extracted using the same methods described in stool DNA extraction (see Section 2.2) and then amplified by PCR using the primer set listed in Table 1 with i-StarTaq DNA polymerase (iNtRON Biotechnology, Gyeonggi, South Korea). The amplification conditions were as follows: pre-denaturation at 95 °C for 5 min; 30 amplification cycle of 95 °C for 15 s, annealing temperature according to Table 1 for 30 s, 72 °C for 1 min; and a final extension at 72 °C for 5 min. The PCR products were purified using NucleoSpin Gel and PCR Clean-up (Macherey-Nagel, Düren, Germany), then cloned using pGEM-T easy vector (Promega, Madison, WI, USA) following the manufacturer’s direction. The plasmid was then transformed to competent cells of *Escherichia coli* strain DH5α using the heat shock method [40] and grown in Luria broth (Difco Laboratories, Detroit, Michigan, USA) supplemented with ampicillin (100 µg/mL) at 37 °C, 120 rpm for 18 h. The plasmid was recovered from *E. coli* DH5α using NucleoSpin Plasmid, Mini kit for plasmid DNA (Macherey-Nagel, Düren, Germany), and then quantified using Nanodrop 2000c (Thermo Scientific, Waltham, MA, USA).

Standard curves for qPCR were generated using SsoAdvanced Universal Inhibitor-Tolerant SYBR Green Supermix (BioRad, Hercules, CA, USA) and SsoAdvanced Universal Probes Supermix (BioRad, Hercules, CA, USA) in LightCycler480 platform (Roche, Basel, Switzerland), according to the kit protocols, with annealing condition following Table 1. The amplifications were performed using duplicate serial dilutions of plasmid DNA. All standard curves used in this study were set to have an efficiency range of 1.8–2 and less than 1% error. The efficiency was calculated using the formula 10^(−1/slope)^, and negative control was included on each run.

#### 2.5.3. Quantification of Targeted Genera in Each Sample by qPCR

Concentration of the DNA template was adjusted to 80–100 ng per reaction. The amplification condition was as follows: pre-denaturation 5 min; 45 amplification cycle of 95 °C for 30 s, annealing temperature according to Table 1 for 20 s and 60 °C for 3 s. For verification purposes, all samples were subjected to melting curve analysis and visualized using agarose electrophoresis. Reactions with specific target band size (according to electrophoresis) and a threshold crossing point within the standard curve range were used for further analysis. Copy numbers of targeted fungi genera for each sample were then determined and converted to copy numbers per gram stool.

### 2.6. Meta-Proteomics Analysis of Five Distinct Genera

#### 2.6.1. Sample Preparation

Stool samples were homogenized with 50 mM phosphate buffer (pH 7), and the protein extraction was carried out following the techniques previously described [41]. The total soluble protein concentration of each extraction was determined using the folin–phenol reagent [42]. Bovine serum albumin (BSA) was used as the standard for the measurement. Five micrograms of protein were reduced using 5 mM dithiothreitol in 10 mM ammonium bicarbonate. The sample was incubated at 60 °C for 1 h, with the alkylation process performed using 15 mM of iodoacetamide in 10 mM ammonium bicarbonate at room temperature for 45 min without light. Samples were then mixed with sequencing grade trypsin (1:20) (Promega, Madison, WI, USA) and incubated at 37 °C overnight. The digested samples were dried and kept at −20 °C.

#### 2.6.2. Liquid Chromatography–Tandem Mass Spectrophotometry (LC–MS/MS)

Tryptic digested peptide samples were resuspended in 0.1% formic acid. Each sample was injected three times into the Ultimate^TM^ 3000 Nano/Capillary LC System (Thermo Scientific, Waltham, MA, USA) coupled to a Hybrid Quadrupole Q-TOF impact IITM (Bruker Daltonics, Billerica, MA, USA) equipped with a nano-captive spray ionization source. Briefly, 100 ng of peptide digests were enriched on a µ-Precolumn 300 µm i.d. × 5 mm C18 Pepmap 100, 5 µm, 100 A (Thermo Scientific, Leicestershire, UK), separated on a 75 μm i.d. × 15 cm and packed with Acclaim PepMap RSLC C18, 2 μm, 100Å, nanoViper (Thermo Scientific, Leicestershire, UK). The reverse phase C18 column was enclosed in a thermostatted column oven set to 60 °C. Solvents A and B containing 0.1% formic acid in water and 0.1% formic acid in 80% acetonitrile, respectively, were used. The peptides were eluted by a gradient of 5–55% solvent B at a constant flow rate of 0.30 μL/min for 30 min. Electrospray ionization was carried out at 1.6 kV, with nitrogen used as the drying gas at a flow rate of 50 L/h. Collision-induced-dissociation (CID) product ion mass spectra were obtained using nitrogen as the collision gas. Mass spectra (MS) and MS/MS spectra were obtained in the positive-ion mode at 2 Hz over the range (*m*/*z*) 150–2200. The collision energy was adjusted to 10 eV as a function of the *m*/*z* value. The LC–MS analysis of each sample was conducted in triplicate. The protein spectral data used in this study have been deposited at ProteomeXchange: PXD027687 and JPST001281.

#### 2.6.3. Quantification of Detected Protein Using Bioinformatics Tools

Proteins in individual samples were bioinformatically quantified by MaxQuant 1.6.6.0 using the Andromeda search engine to correlate MS/MS spectra to the *Acremonium* sp., *Kodamaea* sp., *Rhizopus* sp., *Rhodotorula* sp., and *Wickerhamomyces* sp. protein sequences from the Uniprot database [43]. Label-free quantitation with MaxQuant’s standard settings was performed using a maximum of two miss cleavages, mass tolerance of 0.6 Dalton for the main search, trypsin as the digesting enzyme, carbamidomethylation of cysteine as fixed modification, and oxidation of methionine and acetylation of the protein N-terminus as variable modifications. Peptides with a minimum of seven amino acids and at least one unique peptide were considered for protein identification and used for further data analysis. The protein FDR was set at 1%. The maximum number of modifications per peptide was set to 5. Peptides with maximum intensity from three injections were detected as spectral data of the total proteins expressed from each genus. Only proteins found in at least 50% of the total samples in each group were log2 transformed and used for further analysis.

#### 2.6.4. Determination of Meta-Proteomics Activity of Five Distinct Genera

The log2 transformed proteins were grouped according to the genera to observe the protein abundance profile in each infant. Total protein intensity from each group was then subjected to statistical analysis to determine the differentially expressed proteins (DEP).

Determination of the unique proteins from each group was performed using jvenn viewer [44]. Proteins that only belonged to one group were considered unique and used for the functional category study by running the unique protein sequences against the KEGG database using GhostKOALA [45].

### 2.7. Statistical Analysis

Data for statistical analysis were checked for distribution using the normality test of Shapiro–Wilk in Graph Pad Prism 8.4.3. The method of statistical analysis was decided based on the normality test result. All groups considered as parametric were analyzed by *t*-test (for 2 groups) and ANOVA (for 3 groups or more). For non-parametric data, the Mann–Whitney test (for 2 groups) and the Kruskal–Wallis test (for 3 groups or more) were used. All parameters with at least three group comparisons were subjected to post hoc analysis. Correction for multiple comparison analysis was performed by controlling the FDR using the two-stage step-up method of the Benjamini, Krieger, and Yekutieli algorithm [46].

## 3. Results

### 3.1. Relation of Infant Characteristics

Thirty-four infants (healthy *n* = 17, outgrown *n* = 7, and persisted *n* = 10) were included in the study. Statistical analyses of clinical data showed no significant differences (*p* > 0.05) between each parameter that might interfere with data observation (Table 2).

### 3.2. Diversity of Infant Gut Mycobiome

The sequencing yielded a total of 1,881,148 sequence reads, ranging from 25,872 to 96,180 reads with 55,327 average reads per sample. The fungal alpha diversity was assessed in each group (Figure 1A). Shannon index values indicating the richness and evenness of taxa in a community altered during atopic dermatitis manifestation. In the healthy and outgrown groups, the Shannon index had median values of 1.88 and 1.81, respectively. The median value for the persisted group was 2.16, indicating higher fungal alpha diversity compared to the other groups. To clarify which aspects had higher impacts on the Shannon index value, the richness (Chao1) and Pielou’s evenness index were also investigated. The fungi richness was not related to atopic dermatitis, indicated by the median value of the Chao1 index ranging from 152 to 157.8 in all groups, whereas the evenness of the persisted group was slightly higher (Pielou’s index = 0.42) compared to outgrown (0.34) and healthy (0.36).

The beta diversity was analyzed using the Bray–Curtis dissimilarity resemblance analysis to the relative abundance data of each taxon at class level. Principal component analysis of the beta diversity revealed no clear separation between the three groups (Figure 1B). The clustering analysis indicated that the mycobiome in the outgrown group had a closer resemblance with both healthy and persisted groups.

### 3.3. Distinctive Gut Mycobiome Profile between Healthy, Outgrown, and Persisted Infant Groups

The majority of the sequences (>90%) were successfully classified to the phylum level. The two major phyla *Ascomycota* and *Basidiomycota* were detected across all samples, while phylum *Mucoromycota* was present only in 67% of the samples along with some unidentified fungi (Figure 2A). The amounts of these phyla were similar in all groups. Intriguingly, a slightly lower ratio of *Ascomycota* to *Basidiomycota* was observed in the persisted group, ranging from 2.1 to 2.6 times lower than the healthy and outgrown groups (Figure 2B).

At the genus level, *Saccharomyces*, *Aspergillus*, *Candida*, and *Malassezia* were the most common taxa observed in all samples (Figure 3). *Malassezia* was the only predominant fungus from the *Basidiomycota* phyla, whereas the others were from the *Ascomycota* phyla. Statistical analysis was performed to determine the differences between each group at the genus level. Only 72 genera with higher than 0.05% relative abundance were included in the statistical analysis. Results showed that five genera, including *Kodamaea*, *Wickerhamomyces*, *Acremonium*, *Rhodotorula*, and *Rhizopus* were significantly different between groups. In the healthy group, *Kodamaea* and *Wickerhamomyces* were found more frequently (recorded in 59% of samples) with average abundances of 0.1% and 0.13%, respectively. In the outgrown group, *Acremonium* was detected in 100% of the population, with an average abundance of 0.93%, and *Rhizopus* was discovered in 71.4% of the infants, with an average abundance of 0.09%. The genera *Rhodotorula* associated with the persisted group was present in all samples, with an average abundance of 0.4%.

### 3.4. Quantification of Five Target Fungi from Healthy, Outgrown, and Persisted Infant Groups

To obtain absolute quantification of the five significant genera and total fungi, genera-specific primer sets were designed for quantification purposes using the q-PCR technique. The primer set for *Acremonium* failed to amplify due to the diversity of each species and rapid evolution within this genus. Previous research indicated that effective detection of *Acremonium* was possible using a whole sequence of a small subunit combined with a large subunit of rRNA, though the use of q-PCR has made similar approaches impractical [47].

Quantification of each genus was also in agreement with NGS results (Figure 4). The numbers of fungi in all samples were similar. Concentrations of *Kodamaea* and *Wickerhamomyces* were highest in the healthy group (4.38 ± 1.11 and 0.86 ± 1.22 log copy number of 28s rRNA gene, respectively), while *Rhizopus* was highest in the outgrown group (1 ± 1.31 log copy number of 28s rRNA gene), and *Rhodotorula* was highest in the persisted group (2.5 ± 1.15 log copy number of 28s rRNA gene). Overall, a detection limit as low as 0.9–3.2 log copy number of 26s rRNA gene per gram stool was achieved.

### 3.5. Targeted Meta-Proteomics of Significantly Different Genera

A total of 29,447 proteins from five significantly different genera, including *Acremonium* sp., *Wickerhamomyces* sp., *Kodamaea* sp., *Rhodotorula* sp., and *Rhizopus* sp., were acquired based on protein ID from the UniProt database (i.e., 20,846 protein counts from 11 healthy infants, 9670 protein counts from 7 outgrown infants, and 14,768 protein counts from 5 persisted infants). Protein with at least 50% prevalent level in each group was used for further analysis (Table 3). The number of proteins in the persisted group was the highest, followed by healthy and outgrown groups. The average protein per sample in the persisted group was also the highest. *Acremonium* sp., *Wickerhamomyces* sp., *Rhodotorula* sp., and *Rhizopus* sp. in the persisted group were significantly higher than in the healthy and outgrown groups (*p* < 0.05), while numbers of these proteins in the healthy and outgrown groups were similar. Results demonstrated the possibility of alteration in protein production of significantly different genera during atopic dermatitis manifestation.

Analysis of DEP using jvenn viewer (Figure 5) showed 97 unique proteins in the persisted group, with 16 functionally annotated by the KEGG database (Table 4). Most of the proteins were categorized to metabolism (six proteins) and genetic information processing (eight proteins). Notably, proteins related to human disease were found in *Rhodotorula* sp. The RAN binding protein 1 (Rbp1) from *Rhodotorula* sp. was only expressed in the persisted group.

## 4. Discussion

The association between gut mycobiome and atopic dermatitis development was investigated in infants using metagenomics and metaproteomic technology. No significant differences in characteristic demographics were shown between healthy, outgrown, and persisted groups. The persisted group exhibited higher fungal diversity and richness than the healthy and outgrown groups, concurring with Nash et al. [20]. The gut mycobiome in healthy humans has low diversity and is dominated by a few taxa. By contrast, the high diversity of the gut mycobiome was also linked to illnesses including Crohn’s disease and ulcerative colitis by a previous study [48]. Atopic dermatitis in the persisted group was related to a high abundance of mycobiome that reduced after atopic dermatitis development in the outgrown group.

The most prevalent phyla identified in all groups were *Ascomycota* and *Basidiomycota*. Notably, the ratio of *Ascomycota* to *Basidiomycota* in the persisted group was lower than in the healthy and outgrown groups. Previous research reported that patients with fever retinitis, colon cancer, or inflammatory bowel had a ratio of *Ascomycota* to *Basidiomycota* lower than healthy humans [49,50,51]. However, no clear mechanism defining the cause of this occurrence has been discovered.

Results of metagenomic suggest that the healthy, outgrown, and persisted groups contained significant differences in the five fungi genera. *Wickerhamomyces* and *Kodamaea* were found in abundance in the healthy group. In the outgrown group, *Acremonium* and *Rhizopus* were abundant, while *Rhodotorula* was dominant in the persisted group. Members of the *Wickerhamomyes* and *Kodamaea* genera have been identified as promising probiotic and biocontrol agent candidates [52]. Hjortmo et al. (2008) identified yeast from the *Wickerhamomyces* genus as folate producers [53]. Humans are rarely harmed by *Acremonium* and *Rhizopus* species, with frequent *Acremonium* infection found after trauma [54]. *Acremonium* may be able to synthesize the antibiotic cephalosporin, which is effective against Gram-positive bacteria. *Rhizopus* is a type of foodborne mold that is commonly utilized in the production of fermented foods in Southeast Asia. Some species of this genus may act as pathogens in immunocompromised individuals [55]. *Rhodotorula* is commonly associated with skin barrier disruption [56], and its presence in immunocompromised individuals is often severe [57]. In a recent study, risks caused by the blooming of the *Rhodotorula* genus were addressed, highlighting the increased risk posed in immune-challenged individuals and infants [58].

The fungal numbers in the three sample groups were comparable; however, protein levels in *Acremonium*, *Wickerhamomyces*, *Rhodotorula*, and *Rhizopus* were substantially higher in the persisted group than in the outgrown and healthy groups (*p* < 0.05). This result suggested that fungi were not only present but also metabolically active in the guts of infants, emphasizing the link between gut mycobiome and host health [59,60].

A high abundance of fungal proteins in the persisted group was observed in *Rhodotorula* sp., and three functions were categorized as metabolism, the genetic information process, and human disease. RAN-binding protein 1 (Rbp1) is a human disease-linked protein. In a recent humoral study in mice by Buldain et al. [61], Rbp1 from *Lomentospora* was shown for the first time to act as a fungal allergen. The allergic reaction occurred due to the recognition of immunoglobulin G (Ig-G) toward Rbp1 [62]. Despite the lack of evidence, *Rhodotorula* sp. Rbp1 may act as an allergen with recognition by Ig-G. Although Ig-G is less commonly linked to atopy allergy than Ig-E, Ig-G levels in the blood were higher in atopic dermatitis patients on many occasions [63,64]. This finding suggested an unconventional way of atopic dermatitis development through an Ig-G mediated pathway.

Glycerol kinase from *Rhodotorula* sp. was the unique protein in the persisted group. The glycerol kinase is involved and crucial during triacylglycerol biosynthesis. Overexpression of glycerol kinase leads to higher levels of triacylglycerol and the intermediate products in the related pathway. Bublin et al. [65] reported that lipids from fungi and bacteria could activate immunological responses. The ability of lipids to modify allergen intakes through the intestinal barrier is an intriguing aspect of their role in the induction of allergic responses. *Rhodotorula* has also demonstrated the capability of paracellular translocation from the gut cell wall to the bloodstream, causing sepsis and activating immunological responses [66].

A recent functional metabolic analysis based on a proteomics investigation supported *Rhodotorula*’s involvement in atopic dermatitis. *Rhodotorula* has disseminating properties with unique cell walls, and Rbp1 protein expression, in combination with an underdeveloped gastrointestinal barrier, may play a major role in atopic dermatitis manifestation.

## 5. Conclusions

Our findings showed dysbiosis of the gut mycobiota both during and after the development of atopic dermatitis. *Rhodotorula* overgrowth was a distinguishing characteristic associated with the persisted group. Future investigations are required to examine the related functional roles of *Rhodotorula* in infants with atopic dermatitis.

## Figures and Tables

**Figure 1 jof-07-00748-f001:**
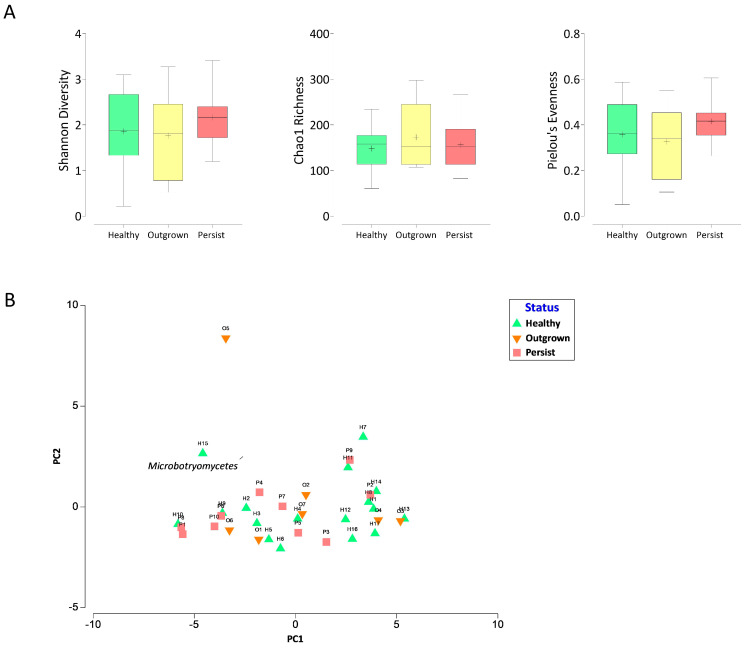
Gut mycobiome alpha diversity (**A**) and beta diversity (**B**) in healthy and atopic dermatitis infants. The alpha diversity is depicted as box and whisker plots; the horizontal line represents the median value, while the ‘+’ represents the mean value. Beta diversity is depicted by a Principal Component Analysis (PCA) plot generated based on the relative abundance percentage of fungi at the taxonomic class level. Bi-plot analysis was used to plot significantly different abundant taxa (*p*-value 0.05). There was no statistically significant difference between the three groups (*p* > 0.05).

**Figure 2 jof-07-00748-f002:**
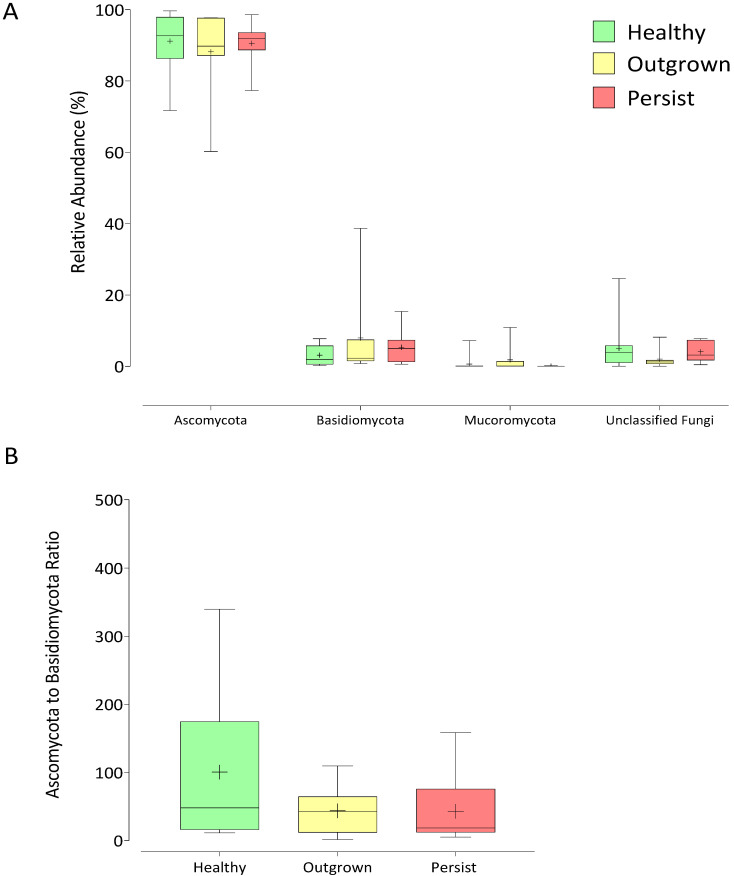
Gut mycobiome of healthy infants and infants with atopic dermatitis. Box and whisker plots depict fungal diversity classification at phylum level (**A**) and the ratio of *Ascomycota* to *Basidiomycota* phyla in each group of infants (**B**). The horizontal line represents median value, while the ‘+’ represents mean value. There was no statistically significant difference between the three groups (*p* > 0.05).

**Figure 3 jof-07-00748-f003:**
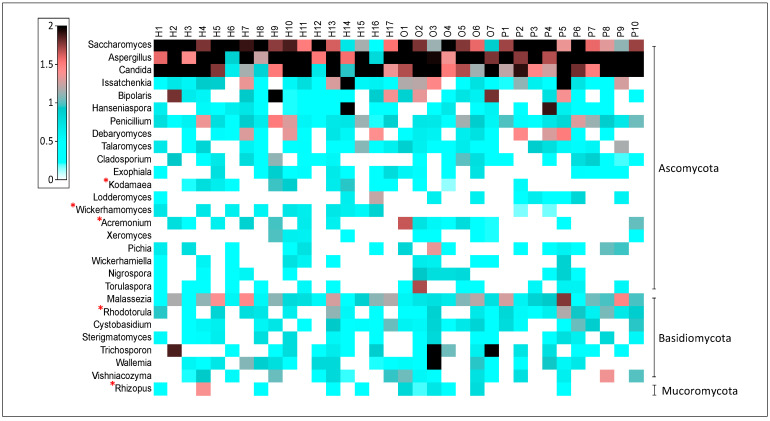
The relative abundance of fungal genera and prevalent compromising the gut mycobiome of healthy and atopic dermatitis infants. The heat map portraying the relative abundance (root four value) of each taxon per infant (H—healthy, O—outgrown, P—persisted). The genera marked with a red asterisk differed significantly between the three groups (*p*-value 0.05).

**Figure 4 jof-07-00748-f004:**
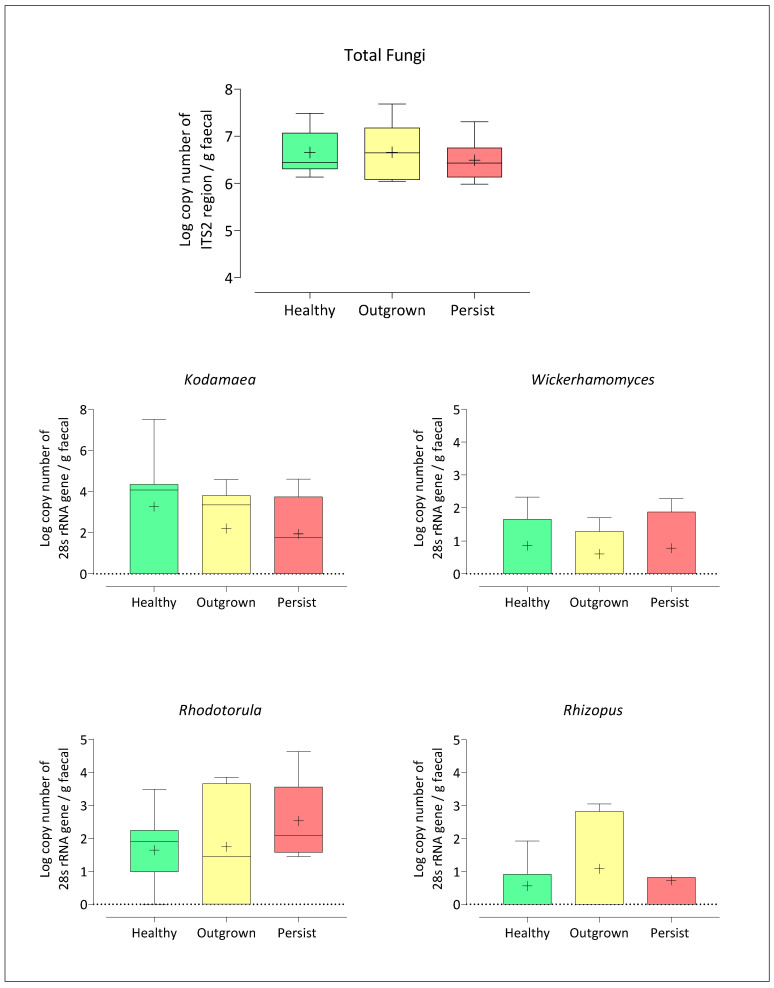
Quantification of altered total fungi and genera, including *Kodamaea* sp., *Wickerhamomyces* sp., *Rhodotorula* sp., and *Rhizopus* sp. The horizontal line represents the median value, while ‘+’ represents the mean value. There was no statistically significant difference between the three groups (*p* > 0.05).

**Figure 5 jof-07-00748-f005:**
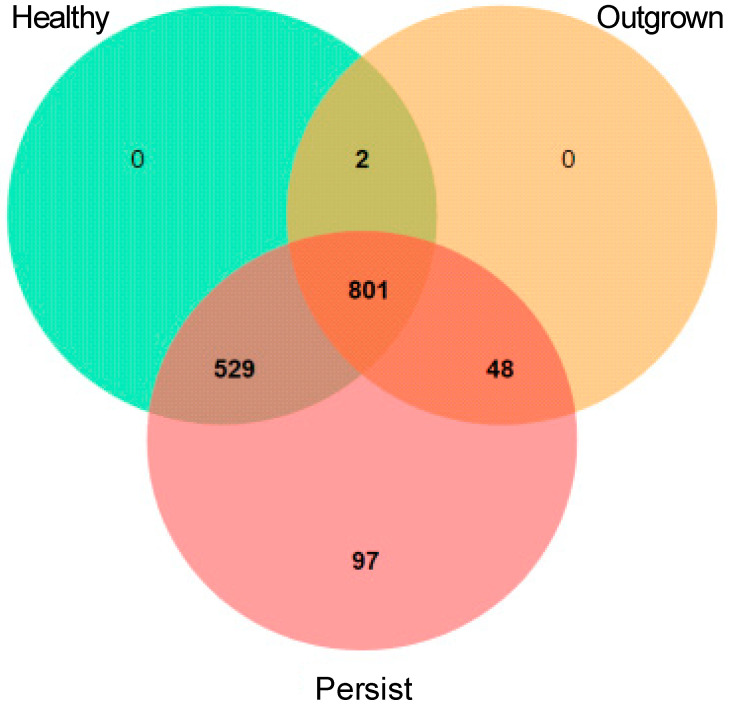
Venn diagram depicting the number of proteins from *Rhodotorula* sp. in each group of infants.

**Table 1 jof-07-00748-t001:** List of primers and probe sequences for quantitative real-time polymerase chain reaction.

Assay	Primer Sequence (5′ to 3′)	Annealing Temp (°C)	Size (bp)	Reference
Total fungi	ITS3: GCATCGATGAAGAACGCAGC ITS4-NGS: TCCTCCGCTTATTGATATGC	56	Variable	[2]
*Rhodotorula* sp.	Rho-F: TCCGGGATGGATAATGGTAGAGAG Rho-R: CCTAAGTGTGAAGAGGTCGAAAC	54	129	This study
*Wickerhamomyces* sp.	Wic-F: TGGGTTTGGCTCTTGTCTATG Wic-R: CGACTCTTCGATAGCACCTTAC	54	104	This study
*Rhizopus* sp.	Rz-F: CTGTAGAAGTGTTTTCCAGGCAAGCCG Rz-R: CCCAAACAACCTGACTCTT	56	133	This study
*Kodamaea* sp.	Kod-F: CGACCGTGTCTATGTTCCTTGGAAC Kod-R: ACTCGTCGAGAGCGCCTTAT	56	95	This study
*Acremonium* sp.	Acre-F: GAGACCTTAACCTACTAACTAGCCCGTAT Acre-R: GTTATTGCCTCAAACTTCCCTCGGCTTG Probe: GCTTTGGCAGTACGCTGGCTTCTTAG	54	98	This study

**Table 2 jof-07-00748-t002:** Clinical characteristics of infants.

Parameter	Healthy	Outgrown	Persisted	*p*-Value
Sampling point (month)	10.59 ± 1.54	10.29 ± 1.6	10.2 ± 1.55	0.79
Sex (female/male)	6/11	3/4	3/7	0.7581
Gestational age (month)	38.45 ± 2.02	39.14 ± 1.86	38.7 ± 2	0.5167
Birth delivery mode (v/cs)	12/5	7/0	5/5	0.0957
Birth weight (g)	3207 ± 215.2	3057 ± 239.5	2967 ± 411.1	0.2113
Breastfeed duration (month)	8.96 ± 4.27	6.71 ± 5.02	7.85 ± 4.44	0.4723
Age of solid food introduction (month)	5.64 ± 0.92	5.43 ± 0.98	5.4 ± 0.84	0.7512
Severity of atopic dermatitis (mild/moderate)	−	2/7	2/10	0.4593

v—vaginal, cs—caesarian section. Values are based on average ± standard deviation.

**Table 3 jof-07-00748-t003:** Number of proteins from five fungi with more than 50% prevalence level per group.

Fungi	Healthy (*n* = 11)	Outgrown (*n* = 7)	Persisted (*n* = 5)	*p*-Value
Total protein	1341	858	1485	
Average protein per sample	122	143	297	
*Acremonium* sp.	339.6 ± 281 ^b^	160.9 ± 261.5 ^b^	889.5 ± 133.2 ^a^	0.0023
*Kodamaea* sp.	20.21 ± 13.28 ^a^	10.64 ± 12.26 ^a^	30.42 ± 3.43 ^a^	0.0784
*Wickerhamomyces* sp.	355.5 ± 294.5 ^b^	168 ± 270.9 ^b^	956.4 ± 151.4 ^a^	0.0021
*Rhodotorula* sp.	368.9 ± 299.7 ^b^	178.5 ± 292.5 ^b^	971.5 ± 143.6 ^a^	0.0026
*Rhizopus* sp.	351.4 ± 287 ^b^	162.4 ± 268.4 ^b^	940.8 ± 138.8 ^a^	0.0022

^a,b^ Significance between sample groups is indicated by different lowercase superscripts (*p* < 0.05). Values are based on average ± standard deviation.

**Table 4 jof-07-00748-t004:** Functional categories of the unique proteins in the persisted group.

Accession Number	Organism	Functional Category	Sub-Functional Category	Protein Name
A0A086TGW8	*Acremonium* sp.	Metabolism	Amino acid metabolism	4-hydroxyphenylpyruvate dioxygenase
A0A086T8N3	*Acremonium* sp.	Metabolism	Carbohydrate metabolism	1,3-beta-glucan synthase
A0A086SWD1	*Acremonium* sp.	Organismal systems	Endocrine system	exocyst complex component 7
A0A086TCH4	*Acremonium* sp.	Genetic information processing	Folding, sorting, and degradation	ubiquitin-conjugating enzyme E2
A0A086TBI1	*Acremonium* sp.	Metabolism	Lipid metabolism	glycerophosphodiester phosphodiesterase
A0A086T3W0	*Acremonium* sp.	Metabolism	Metabolism of cofactors and vitamins	5-formyltetrahydrofolate cyclo-ligase
A0A086TD10	*Acremonium* sp.	Genetic information processing	Replication and repair	bloom syndrome protein
A0A367JPM9	*Rhizopus* sp.	Metabolism	Amino acid metabolism	gamma-glutamyl hercynylcysteine S-oxide synthase
A0A2G4SN35	*Rhizopus* sp.	Genetic information processing	Folding, sorting, and degradation	20S proteasome subunit alpha 4
I1CUB3	*Rhizopus* sp.	Genetic information processing	Translation	small subunit ribosomal protein
A0A0K3CQU6	*Rhodotorula* sp.	Metabolism	Lipid metabolism	glycerol kinase
A0A511KMS2	*Rhodotorula* sp.	Genetic information processing	Translation	H/ACA ribonucleoprotein complex subunit 4
A0A511K989	*Rhodotorula* sp.	Human disease	Cancer	RAN-binding protein 1
K0KHA2	*Wickerhamomyces* sp.	Genetic information processing	Folding, sorting, and degradation	YidC/Oxa1 family membrane protein insertase
A0A1E3NZ77	*Wickerhamomyces* sp.	Genetic information processing	Replication and repair	DNA excision repair protein ERCC-6
K0KHS6	*Wickerhamomyces* sp.	Genetic information processing	Transcription	DNA-directed RNA polymerase I subunit RPA2

## Data Availability

The raw Illumina sequences of ITS2 metagenome used in this study have been deposited at the NCBI shorts read archive (SRA) with the BioProject accession number PRJNA751473. The protein spectral data used in this study have been deposited at ProteomeXchange: PXD027687 and JPST001281.

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
