# Peer review of "ITS2 Sequencing and Targeted Meta-Proteomics of Infant Gut Mycobiome Reveal the Functional Role of Rhodotorula sp. during Atopic Dermatitis Manifestation"

_jof, 2021, doi:10.3390/jof7090748_

Round 1

Reviewer 1 Report

Overall, the study aim to study the causal relationship between fungi and atopic dermatitis in infants, however the study can not address this key question.

After reviewing the manuscript, I may raise comments as below and suggest a minor revision:

The manuscript is well written. I have major concerns as below:

  1. The authors aim to study the causal relationship however by a cross-sectional nature, this study cannot address such relationship.
  2. In figures 1 2 4, statistical significance should be labeled
  3. In table 4, protein expression level is not meaningful

Author Response

Point 1: Overall, the study aims to study the causal relationship between fungi and atopic dermatitis in infants, however the study cannot address this key question. After reviewing the manuscript, I may raise comments as below and suggest a minor revision:

Response 1: Thank you for taking the time to read this manuscript; we will respond to your comments and concerns for each item below.

Point 2: The authors aim to study the causal relationship however by a cross-sectional nature, this study cannot address such relationship. 

Response 2: We agree with your comment. The objective was revised to “the gut mycobiome (fungal microbiome) of infants with atopic dermatitis was investigated using both metagenomic and metaproteomic approaches to capture mycobiome status links to the disease” as shown in line 85-86

Point 3: In figures 1 2 4, statistical significance should be labeled

Response 3: Thank you for your view. However, these data from figures 1, 2 and 4 were not statistically significant different between each group. For minimize the misunderstanding we already add a sentence “There was no statistically significant difference between the three groups (p > 0.05)” in figure legend.

Point 4: In table 4, protein expression level is not meaningful

Response 4: We agreed with your comment. Protein expression level was removed from the table.

Reviewer 2 Report

interesting paper on the role of (intestinal) fungi in atopic dermatitis, in which both the comparison of a persist and outgrown group and the use of metaproteomics adds new elements. In certain respects it looks like the healthy and persist groups are more comparable than the outgrowth group (e.g. in the proteomics data). This aspect might get some more attention the discussion. Furthermore a small spelling check would identify some typo's 

Author Response

Point 1: Interesting paper on the role of (intestinal) fungi in atopic dermatitis, in which both the comparison of a persist and outgrown group and the use of meta-proteomics adds new elements. In certain respect it looks like the healthy and persist groups are more comparable than the outgrowth group (e.g., in the proteomics data). This aspect might get some more attention the discussion. Furthermore, a small spelling check would identify some typo's

Response 1: Thank you so much for your kind comment and suggestion. In the part of metaproteomic, the number of proteins in outgrown and healthy were similar. Only persist group found the unique proteins that different from control and outgrown. That’s the reason why we focus the protein from persist. In addition, after we do a statistical analysis of the total protein and genera specific protein between the three groups, we found that persist groups differ significantly from the healthy and outgrown group (Table 3). We already rechecked the spelling as well.

Reviewer 3 Report

The work is relevant, performed qualitatively, written interestingly, the connection between the development and features of the course of atopic dermatitis in children with fungal colonization is proved. The work is promising for further study of the mechanisms of participation specific fungus  (Rhodotorula) in the pathological process - in order to find new diagnostic and therapeutic approaches to this pathology.

There are several suggestions to the authors - to make a small edit to make it easier to understand the results. 

For example,

1)  in Table 3 there is no footnote - what do a) and b) mean

2)  in Table 3 replace the brackets in the line "Total protein )average protein
per sample (  "   1341 )122(    858 )143(      1485 )297( 

3) in Table 3 the data are presented for comparing three groups of patients, but only one p-value (?)

4) In the Supplement, I would like to have a transcript of the abbreviations GA, CS, BW, AD, etc

Author Response

Point 1: The work is relevant, performed qualitatively, written interestingly, the connection between the development and features of the course of atopic dermatitis in children with fungal colonization is proved. The work is promising for further study of the mechanisms of participation specific fungus (Rhodotorula) in the pathological process - in order to find new diagnostic and therapeutic approaches to this pathology.

Response 1: We appreciate your positive feedback and interest in our study.

Point 2: In Table 3 there is no footnote - what do a) and b) mean

Response 2: We already added the footnote “a,b Significance between sample groups are indicated by different lowercase superscripts (p≤0.05)”

Point 3: in Table 3 replace the brackets in the line "Total protein) average protein per sample (  "   1341 )122(    858 )143(      1485 )297(

Response 3: The average protein per sample were removed from bracket and added to new line in Table 3.

Point 4: in Table 3 the data are presented for comparing three groups of patients, but only one p-value (?)

Response 4: We apologised for unclear information. The p value shown in Table 3 were calculated from Kruskal-Wallis test that compared across three groups. For comparison between two subgroups, the p value<0.05 are indicated by different lowercase superscripts. We already added the information in Table legend.

Point 5: In the Supplement, I would like to have a transcript of the abbreviations GA, CS, BW, AD, etc

Response 5: Thank you very much for your view. We have provided the abbreviations for additional files 1.
